# Estimating access to surgical care: A community centered national household survey from Pakistan

**Muhammad Nabeel Ashraf**[1,2], **Irum Fatima**[3], **Ammar Ali Muhammad**[3], **Katherine Albutt**[4], **Manon Pigeolet**[5,6], **Asad Latif**[7], **John G. Meara**[5,8], **Lubna Samad**[9]*

1 Indus Hospital and Health Network, Karachi, Pakistan, 2 Medical College of Georgia, Augusta University, Augusta, GA, United States of America, 3 Interactive Research and Development (IRD), Karachi, Pakistan, 4 MGH, Boston, MA, United States of America, 5 Program in Global Surgery and Social Change, Harvard Medical School, Boston, MA, United States of America, 6 Université Libre de Bruxelles, Faculty of Medicine, Brussels, Belgium, 7 Department of Anaesthesia, Aga Khan University, Karachi, Pakistan, 8 Department of Plastic and Oral Surgery, Boston Children's Hospital, Boston, MA, United States of America, 9 Interactive Research and Development (IRD) Global, Singapore, Singapore

* lubna.samad@ird.global

**Data Availability Statement:** The datasets generated and/or analyzed during the current study are available at https://dataverse.harvard.edu/file.xhtml?fileId=7061321&version=DRAFT.

## Abstract

Pakistan is a lower middle-income country in South Asia with a population of 225 million. No estimate for surgical care access exists for the country. We postulate the estimated access to surgical care is less than the minimum 80% to be achieved by 2030. We conducted a randomized, stratified two-stage cluster household survey. A sample of 770 households was selected using 2017 census frames from the Pakistan Bureau of Statistics. Data was collected on choice of hospital and travel time to the chosen hospital for C-section, laparotomy, open fracture repair (OFR), and specialized surgery. Analysis was conducted using Stata 14. Access to all Bellwether surgeries (C-section, laparotomy, and open fracture repair) in Pakistan is estimated to be 74.8%. However, estimated access in rural areas and the provinces of Balochistan, Khyber Pakhtunkhwa (KP) and Sindh is far less than in urban areas and in Punjab and Islamabad. Estimated access to C-sections is more compared to OFR, laparotomy, and specialized surgery. Health system strengthening efforts should focus on improving surgical care access in rural areas and in Balochistan, KP, and Sindh. More focus is required on standardizing the availability and quality of surgical services in secondary-level hospitals.

## Introduction

Surgical and obstetric diseases contribute significantly to the global disease burden. Shrime et al., 2015 reports health care providers estimate 28–32% of the global disease burden is attributable to surgical care diseases [1]. Much of this disease burden is preventable and is in low- and middle-income countries (LMIC) [2]. Bickler et al., 2015 used 2010 Global Burden of Disease (GBD) data and found 1.4 million deaths and 77.2 million Disability Adjusted Life Years (DALYs) in LMICs occur due to a lack of essential and emergency surgical care at primary,

**Funding:** This work was supported by the Harvard Medical School Center for Global Health Delivery - Dubai (granted to JGM and LS). MNA, IF, LS received partial salary support services from this grant. Belgian Kids' Fund for Pediatric Research (granted to MP) was used by MP for scholarly support. The funders had no role in study design, data collection and analysis, decision to publish, or preparation of the manuscript.

**Competing interests:** The authors have declared that no competing interests exist.

secondary and tertiary levels of healthcare. Injuries (77%), maternal and neonatal conditions (14%), and gastrointestinal conditions (9%) contributed to the majority of preventable mortality [3]. South Asia and Sub-Saharan Africa have the highest percentage of preventable deaths among the seven LMIC supra-regions defined by GBD [3]. Other than essential and emergency surgeries, elective and non-urgent specialized surgical care for cataracts, cleft lip and palate, congenital heart diseases, neural tube defects and obstetric fistula in LMIC results in 388,000 deaths and 38.9 million DALYs [3].

Access to surgical care is one indicator that can help gauge and guide the development of a health system's capacity to address the surgical disease burden [2]. A recent analysis of maternal mortality showed an inverse relation between mothers who receive timely emergency obstetric care and maternal mortality ratio [4]. In 2015, The Lancet Commission on Global Surgery (LCoGS) estimated 5 billion people lack access to timely, safe, quality, and affordable surgical care globally and recommends each country should aim to provide 2-hour geographical access to a Bellwether hospital for 80% of the population [2]. Bellwether hospitals are defined as facilities performing laparotomy, cesarean sections (C-section), and treatment of open fractures. The capacity to perform these three procedures (Bellwether procedures) is associated with the capacity to provide a wider range of emergency operations as listed by the World Health Organization (WHO) [5].

Pakistan is a lower middle-income country in South Asia with a total population of 225 million people and spread over 800,000 square kilometers [6]. The National Vision of Surgical Care, Pakistan's guiding document for National Surgical Obstetric and Anesthesia Planning (NSOAP), recommends all provinces should aim to achieve 2-hour access for 80% of the population. No estimate for surgical care access is available in Pakistan, but based on available research, we postulate population access to surgical care is poor [7]. We used a novel community-based approach and conducted a 2 stage randomized cluster household survey to estimate access to surgical care and identify disparities between rural/urban, provinces, and household consumption quintiles. Our study relied on community responses and knowledge about the locally preferred functional surgical facility and local transport times which we believe provide a better estimate of surgical care compared to GIS-based approach which ignores disparity of road infrastructure and surgical facility functionality. We believe this study provides information to develop targeted health-system-strengthening interventions. Our unique and novel experience of measuring access through a household survey provides vital lessons that can guide better data collection for surgical care access in the future.

## Methods

### Study timeline and setting

This cross-sectional study was conducted from January 2019 to November 2020, with an interruption in field activities due to the Covid-19 pandemic from April 2020 to September 2020. Data was collected across the country. Pakistan is a federation with four federating units called provinces: Balochistan, Khyber Pakhtunkhwa (KP), Punjab, and Sindh; a federal capital territory, Islamabad; and two autonomous areas: Azad Jammu Kashmir (AJK) and Gilgit Baltistan (GB) [8]. Due to the unavailability of the population data frames for AJK and GB, these areas were not included in the survey.

### Sample size and study sites

A stratified two-stage cluster sampling methodology was used to get a nationally representative sample. An effective sample size of 285 was calculated using the formula n = [Np(1-p)]/ [(d2/

Z21-α/2*(N-1)+p*(1-p)], assuming p, the estimated prevalence of untreated surgical disease as 0.25 based on multinational and regional studies [9, 10], and a level of significance of 0.05. We assumed access to surgical care is spatially correlated, therefore a higher than normal intra-cluster cluster coefficient (ICC) of 0.167 was used similar to ICC recommended for immunization coverage surveys by WHO [11]. The target number of respondents per cluster (m) was taken as 7 based on our operational capacity. The design effect was therefore calculated to be 2 (DEFF = 1 + (m− 1) * ICC). Accounting for a 75% response rate, we calculated a final sample size of 770, leading to a need for 110 clusters.

Clusters were allocated to provinces proportional to their household populations. Six out of the 110 (5.5%) clusters were allocated to Balochistan, 15 (13.7%) were allocated to KP 58 (53.1%) to Punjab, 29 (26.7%) to Sindh, and 2 (1%) clusters were allocated to Islamabad Capital Territory (ICT).

Tehsils, the third degree administrative divisions in each province, were used as primary sampling units while households were used as secondary sampling units (PSUs and SSUs, respectively) [8, 12]. The tehsil and household sampling frames were obtained from the Pakistan Bureau of Statistics (PBS), and we systematically selected our sample across Pakistan. Some PSUs were sampled twice, in which case 14 instead of 7 households were sampled in the tehsil. For KP, one of the selected tehsils was replaced due to security concerns. A tehsil of similar characteristics was selected. A map of final sample sites is shown in Fig 1.

## Interview tool

The interview tool was adapted from the Surgeons Overseas Assessment of Surgical Needs (SOSAS) tool [13]. The tool included sections on household enumeration, socioeconomic and demographic characteristics, need and accessibility for C-section, laparotomy, open fracture repair (OFR), and other specialized surgery, deaths in the family and verbal autopsy for reported deaths. The concept of OFR expanded upon the original type of fracture repair included in the Bellwether definition and hence we included all fractures that underwent treatment through a surgical incision, irrespective of whether these fractures were open or closed fractures. This decision was taken out of practical consideration for the interviewers to avoid confusion about what constitutes an open fracture and to measure access to definitive fracture instead of urgent treatment of open fracture. Procedures were described to the respondents in their local language using lay people terms. Rather than determining 2-hour access using a dichotomous question, the respondents were asked about the hospital they would go to for the particular procedure, what transport they would use to travel, and based on the transport used, how long it would take to reach the preferred facility. We also added questions regarding household consumption from the PGSSC Financial Risk Protection Survey form [14]. The questionnaire is attached as Annex 1 in S1 Text.

## Data collection

Data collectors from local tehsils were identified, enrolled and trained to administer the study tool. Data collectors visited the households selected, obtained informed consent, and interviewed the head of household. Informed consent was collected on paper by health worker from the head of the household after explaining the risks and benefits of participation in the study in their native language. Data was initially collected on paper forms. The data was then entered into the Redcap online app storing the data to the secure redcap server at the institution [15]. Our study team did not confirm the functionality of facilities reported as preferred facilities by the respondents.

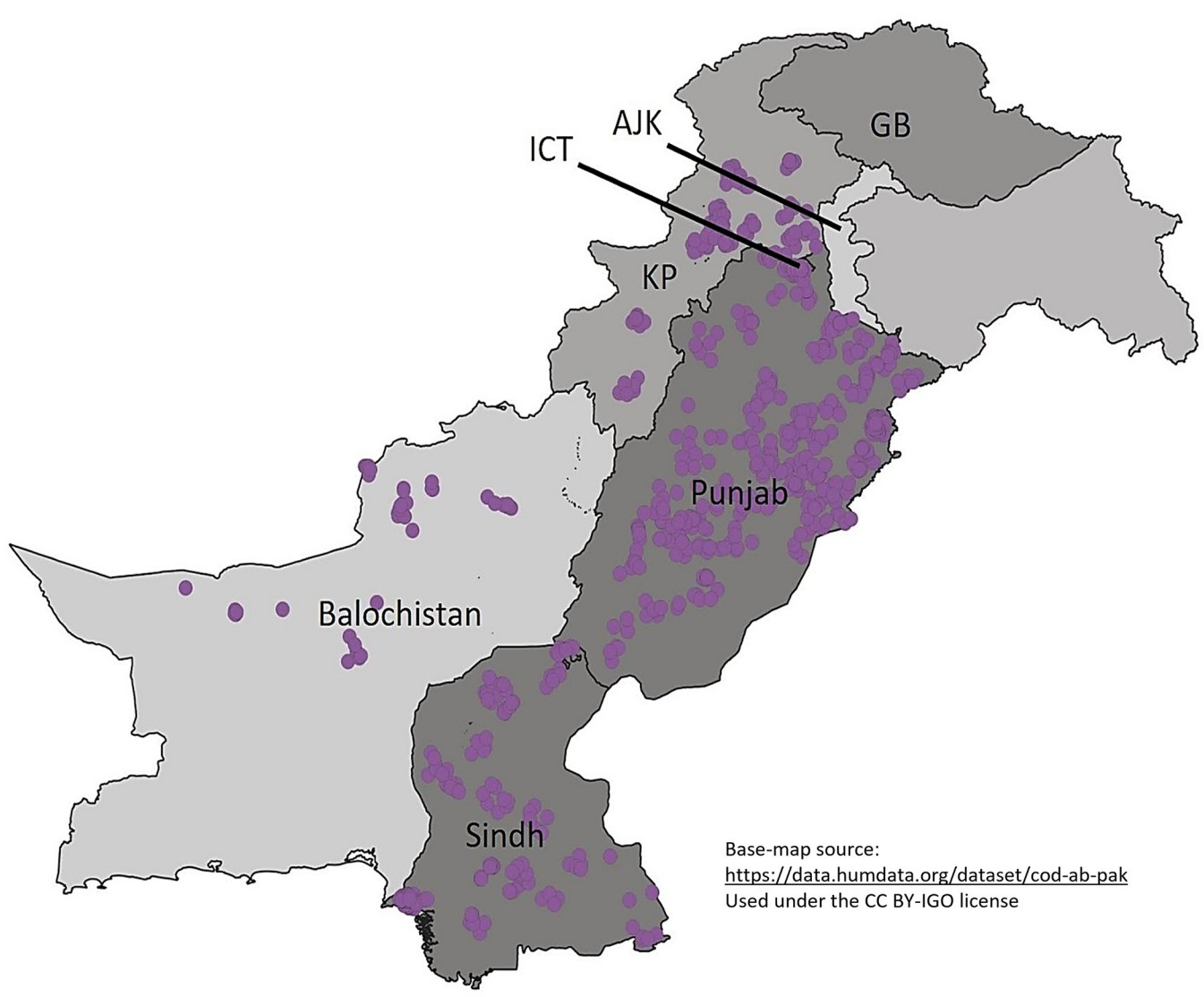

**Fig 1. Map of sample sites across Pakistan.**

## Definitions

**Households with estimated 2-hour access.** Households that reported travel time of less than 2-hours to their preferred surgically capable hospital using their preferred transport.

**Estimated population access to Bellwether procedures.** Proportion of households with reported 2-hour access to all three Bellwether surgeries at the same or different preferred hospital(s).

**Bellwether procedures.** These include Cesarean-section, laparotomy, open fracture repair (OFR).

**Specialized surgery.** Surgery other than Bellwether such as surgery for cancer, plastic surgery, or urological surgery.

## Data analysis

We used STATA 14 to conduct our analysis [16]. For the stratified weighted analysis settings, the stratification variable was provinces. The PSU weight was determined as the number of

households in the tehsil divided by the number of households in the province; the SSU weight was determined as the number of households selected divided by the number of households in the tehsil. The overall weight for each household was determined as a product of PSU and SSU weights. For confidence interval calculation of estimates, the linearized method for variance estimation was used and single sampling units (Islamabad Capital Territory in our case) were treated as a certainty unit.

Accessibility estimates are expressed as weighted proportions of households with reported access. National accessibility estimates for "all surgeries" (C-section, Laparotomy, OFR, and specialized surgery), "Bellwether procedures" (C-section, Laparotomy, and OFR), and individually for "C-section", "Laparotomy", "OFR", and "specialized surgery" are calculated. Subpopulation estimates were calculated for provinces, rural and urban households, and households belonging to the different per capita consumption quintiles. The per capita household consumption categories are the same as per the national household consumption quintiles determined by PBS [17]. Confidence intervals and design effects are also provided.

Results of weighted multiple logistic regression for access across provincial, rural and urban, and national per-capita household consumption quintile sub-populations are also provided.

## Ethical approval

Ethical approval to conduct the study was received from the Interactive Research and Development Institutional Review Board (IRD IRB 2018 05 006) and the Harvard Medical School Institutional Review Board. The Ministry of National Health Service Regulation and Coordination, Government of Pakistan, and the provincial departments of health issued letters of support to conduct the study.

## Results

Out of the 770 households, the provincial distribution was proportional to the population distribution; 53% of the households were from Punjab and the least number of households were from Balochistan (5%) and Islamabad (2%). The proportion of rural households was 56%. Most households (40%) in our sample belonged to the lowest monthly household consumption quintile as per the national standards. Further descriptive statistics are listed in Table 1.

Nationally 74.8% (67.9–80.7%) of households reported they have access to the Bellwether procedures while 62.4% (54.6–69.6%) of households reported access to the Bellwether and specialized surgeries. Estimated population access for C-sections was the highest with 88.7% (84.2–92%) of the households, followed by OFR at 81.9% (76–86.6%), laparotomy at 79% (72.6–84.3%), and specialized surgery was 65.2% (57.4–72.2%) (Table 2).

The estimated population access for Bellwether procedures in rural areas was 64.7% (56.4–72.2%) compared to urban housheolds with 88.2% (81.7–92.6%) (Table 2). After controlling for provincial and household consumption quintile, the odds of access to Bellwether procedures in rural areas compared to access in urban areas is 27.4% (13.3–36.6%) (Table 3). Estimated population access to C-section in rural areas was above 80% while estimated population access to laparotomy, OFR and specialized surgery were all below 80% (Table 2).

Islamabad and Punjab had highest access to Bellwether procedures at 92.9% and 81% (71.9%-87.7%) followed by Sindh at 60.6% (47.6–72.2%), KPK at 48.5% (30.5–66.8%), and Balochistan at 28.6% (10.9–56.8%) (Table 2). After controlling for rural vs urban and household consumption quintiles, the odds for access to Bellwethers in Balochistan, KP, and Sindh, when compared to Punjab, were 12.0%(3.0–46.9%), 27.3% (11.6–64.4%), and 28.8%(14.8–55.7%), respectively (Table 3).

**Table 1. Descriptive statistics for households surveyed.**

| | | | N | % |
|---|---|---|---|---|
| **Average Household Size** | | | 7.0 (±3.4) | |
| **Urbanity** | | | | |
| | Urban | | 338 | 44 |
| | Rural | | 432 | 56 |
| **Province** | | | | |
| | Punjab | | 406 | 53 |
| | Sindh | | 203 | 26 |
| | KP | | 105 | 14 |
| | Balochistan | | 42 | 5 |
| | Islamabad | | 14 | 2 |
| **Household per capita consumption** | | | | |
| | 1st | 1st quintile (upto Rs. 3271) | 304 | 40 |
| | 2nd | 2nd quintile (Rs. 3272–4207) | 122 | 16 |
| | 3rd | 3rd quintile (Rs. 4208–5402) | 101 | 13 |
| | 4th | 4th quintile (Rs.5403 – 7508) | 119 | 16 |
| | 5th | 5th quintile (Rs. 7509 or above) | 124 | 16 |

**Table 2. Overall and stratified estimates for proportion of households with reported access to all procedures, bellwethers, and individual procedures with 95% confidence intervals.** DEFF can be seen in (S1 Table). Proportions are expressed as percentages.

| | | All Procedures (Bellwether+specialized surgery) | Bellwethers | C-section | Laparotomy | OFR | Specialized Surgery |
|---|---|---|---|---|---|---|---|
| *Overall* | | 62.4 (54.6, 69.6) | 74.8 (67.9, 80.7) | 88.7 (84.2, 92.0) | 79.0 (72.6, 84.3) | 81.9 (76.0, 86.6) | 65.2 (57.4, 72.2) |
| *Province* | | | | | | | |
| | Punjab | 69.6 (59.6, 78.1) | 81.0 (71.9, 87.7) | 93.8 (87.6, 97.0) | 82.9 (74.3, 89.0) | 87.0 (79.1, 92.2) | 71.9 (61.9, 80.2) |
| | Sindh | 44.8 (31.0, 59.5.) | 60.6 (47.6, 72.2) | 79.8 (69.5, 87.2) | 72.9 (62.4, 81.4) | 71.9 (60.4, 81.1) | 49.8 (36.0, 63.6) |
| | KPK | 33.2 (19.2, 51.0) | 48.5 (30.5, 66.8) | 58.7 (37.9, 76.8) | 55.6 (34.8, 74.6) | 54.6 (35.1, 72.8) | 34.2 (20.0, 51.9) |
| | Balochistan | 26.2 (9.0, 56.0) | 28.6(10.9, 56.8) | 45.2 (20.6,72.5) | 33.3 (13.4, 61.7) | 40.5 (17.7, 68.3) | 26.2 (9.0,56.0) |
| | Islamabad CT | 92.9 | 92.9 | 92.9 | 92.9 | 92.9 | 92.9 |
| *Urban vs Rural* | | | | | | | |
| | Rural | 51.1 (42.5, 59.7) | 64.7 (56.4, 72.2) | 84.5 (78.3, 89.2) | 69.3 (61.4, 76.1) | 74.1 (66.3, 80.6) | 54.4 (45.7, 62.9) |
| | Urban | 77.3 (69.1, 83.9) | 88.2 (81.7, 92.6) | 94.3 (90.3, 96.7) | 91.9 (86.6, 95.2) | 92.3 (86.7, 95.6) | 79.3 (71.5, 85.4) |
| *Household consumption quintile* | | | | | | | |
| | 1st quintile | 59.3 (49.9, 68.2) | 70.9 (62.3, 78.3) | 86.8 (80.6, 91.2) | 75.7 (67.2, 82.5) | 76.4 (68.3, 82.9) | 62.0 (52.6, 70.6) |
| | 2nd quintile | 56.8 (43.9, 68.8) | 74.4 (60.2, 84.8) | 86.4 (74.1,93.4) | 79.8 (66.7, 88.7) | 80.3 (68.2, 88.6) | 59.2 (46.4, 70.9) |
| | 3rd quintile | 68.3 (55.3, 79.0) | 78.3 (66.0, 87.0) | 91.5 (83.9, 95.7) | 82.3 (70.2, 90.1) | 87.6 (78.4, 93.2) | 70.5 (58.0, 80.5) |
| | 4th quintile | 62.5 (47.6, 75.3) | 77.1 (63.8, 86.6) | 91.9 (85.8, 95.5) | 80.1 (68.0, 88.4) | 84.9 (75.3, 91.2) | 64.2 (49.2, 76.9) |
| | 5th quintile | 70.5 (58.1, 80.5) | 79.4 (69.3, 86.8) | 89.9 (81.2, 94.8) | 82.5 (72.9, 89.2) | 89.2 (80.6, 94.2) | 75.5 (62.7, 84.9) |

**Table 3. Adjusted odds ratio for proportion of households with reported access to all procedures, bellwethers, and individual procedures with 95% confidence intervals.**

| | All Procedures (Bellwether+specialized surgery) | Bellwethers | C-section | Laparotomy | OFR | Specialized Surgery |
|---|---|---|---|---|---|---|
| *Province* | | | | | | |
| Punjab | ref | ref | ref | ref | ref | ref |
| Sindh | 0.29 (0.145,0.579)** | 0.288 (0.148,0.557)*** | 0.242 (0.094,0.62)** | 0.469 (0.242,0.91)* | 0.344 (0.165,0.717)** | 0.322 (0.165,0.627)** |
| KPK | 0.285 (0.115,0.707)** | 0.273 (0.116,0.644)** | 0.114 (0.038,0.343)*** | 0.342 (0.127,0.925)* | 0.212 (0.079,0.567)** | 0.247 (0.104,0.588)** |
| Balochistan | 0.065 (0.017,0.258)*** | 0.12 (0.03,0.469)** | 0.046 (0.011,0.201)*** | 0.069 (0.016,0.293)*** | 0.078 (0.02,0.306)*** | 0.108 (0.028,0.419)** |
| Islamabad CT | 1.318 (0.687,2.526) | 2.912 (1.746,4.857)*** | 0.463 (0.192,1.118) | 1.069 (0.553,2.064) | 0.84 (0.405,1.743) | 2.612 (1.559,4.377)*** |
| *Urban vs Rural* | | | | | | |
| Rural | ref | ref | ref | ref | ref | ref |
| Urban | 0.221 (0.133,0.366)*** | 0.274 (0.179,0.422)*** | 0.332 (0.177,0.621)** | 0.191 (0.117,0.314)*** | 0.238 (0.123,0.462)*** | 0.287 (0.189,0.437)*** |
| *Household consumption quintile* | | | | | | |
| 1st quintile | ref | ref | ref | ref | ref | ref |
| 2nd quintile | 0.985 (0.514,1.889) | 0.723 (0.423,1.237) | 0.826 (0.417,1.635) | 1.074 (0.546,2.114) | 1.083 (0.591,1.985) | 0.725 (0.426,1.233) |
| 3rd quintile | 1.189 (0.618,2.289) | 1.208 (0.658,2.217) | 1.415 (0.722,2.773) | 1.254 (0.583,2.7) | 1.913 (1.009,3.627)* | 1.214 (0.663,2.223) |
| 4th quintile | 0.951 (0.452,1.999) | 0.778 (0.383,1.581) | 1.297 (0.576,2.92) | 0.936 (0.452,1.937) | 1.298 (0.654,2.576) | 0.769 (0.376,1.575) |
| 5th quintile | 1.123 (0.595,2.12) | 1.196 (0.637,2.244) | 1.06 (0.44,2.555) | 1.119 (0.558,2.242) | 2.066 (0.967,4.414) | 1.449 (0.742,2.827) |

*p-value <0.05

**p-value <0.01

***p-value <0.001

Among the 535 households with estimated 2-hour access to Bellwether procedures, 213 (39.8%) reported one facility for all three procedures, 218 (40.8%) reported one facility for 2 of the three Bellwethers, and only 104 (19.4%) reported different hospitals for the Bellwethers. In contrast, among households without 2-hour access, 46 (19.6%) reported one facility for all three procedures, 62 (26.4%) reported one facility for 2 of the three Bellwethers, and 127 (54.0%) reported different facilities for the Bellwethers (Table 4).

## Discussion

Access to Bellwether procedures as estimated by this national household survey is 74.8% and falls short of the 80% benchmark proposed by the LCoGS. The stratified analysis demonstrates the disparity in the distribution of accessibility to surgical care across Pakistan. In rural areas, 65% of the population reported 2-hour access to bellwethers in contrast to 88% in the urban population. A regional difference was noted with more than 80% of reported access

**Table 4. Number of facilities chosen for Bellwether procedure among households with and without two-hour access.**

| | Total (n = 770) | Access (n = 535) | No access (n = 235) |
|---|---|---|---|
| One hospital for all three Bellwethers | 259 (33.6%) | 213 (39.8%) | 46 (19.6%) |
| One hospital for any two procedures | 280 (36.4%) | 218 (40.8%) | 62 (26.4%) |
| Different facilities for the Bellwethers | 231 (30%) | 104 (19.4%) | 127 (54.0%) |

documented only in Punjab and Islamabad Capital Territory. A difference was also noted in access to individual procedures with better estimated population access to C-sections as compared to laparotomy, OFR, and specialized surgery in that order.

Poor surgical care service availability at secondary level hospitals that are supposed to be the front line for emergency surgical services is one of the main reasons for the rural disparity [18]. Despite being designated as centers for essential surgery, surveys of rural secondary-level hospitals have shown that up to 71% do not have an anesthetist, 60% do not have a gynecologist, and equipment for basic surgical care is mostly absent [19, 20]. When available, services are often inadequate and unreliable. Tertiary hospitals, both public and private, located in the urban areas are therefore the major providers of surgery in the country. The majority of their resources are expended in basic surgical care and dealing with emergency, delayed and complicated patients, limiting their capacity to provide specialized surgery.

The suboptimal estimated population access to laparotomy and OFR, as opposed to C-section, indicates a difference in focus among the Bellwether procedures. The wider availability of C-section services is expected, given the international and local focus on maternal health services development. The lack of OFR and laparotomy services exposes Pakistan's population to death and disability from injuries and emergency general surgery conditions, such as intestinal perforation, intestinal obstruction, and appendicitis, which form the bulk of preventable global death and disease burden [3, 18]. Standardizing the service packages and quality requirements at secondary level hospitals to provide laparotomy and OFR services in addition to C-section is therefore extremely important and will lead to more efficient use of resources [5].

The marked provincial disparity is a result of the contextually different logistical, economic, socio-cultural, and political factors. Balochistan with the poorest estimated population access comprises 48% of the total country area but is home to only 6% of the country's population, has a population density of 35 per km square, the lowest financial allocations, the poorest literacy rate at 40%, and has been fraught with security issues that have deterred physician retention and infrastructure development [8, 21–23]. In contrast, Punjab, Sindh, and KP respectively, are home to 53%, 23%, and 17% of the national population. They have population densities of 536 per km square, 340 per km square, and 349 per km square, and literacy rates of 60%, 57%, and 55%. Islamabad is a federally governed small capital territory compared to these provinces and with more structured planning has higher estimated population access [8, 21, 22].

Access to surgical care remains a broad, multi-dimensional concept with interpretation potentially extending from mere geographically available access to cultural, financial, and quality care realized access. Literature review shows surgical care access has been mostly defined as 2-hour access to facilities with Bellwether capacity. The approach to identifying such facilities is only rigorous in some studies. Two-hour access has mostly been estimated based on GIS based car travel time. Many validation studies, however, have shown discordance between GIS calculated and population reported travel time since local road network maps and geographical terrain variability are not adequately documented in LMICs [24].

Several methods of 2-hour access estimation have been proposed in the literature predominantly using some form of GIS estimates [25–31]. Compared to these studies, our approach to estimating surgical care access is novel and unique. We decided a GIS study would not provide a comprehensive and accurate estimate of the population's access to surgical care in Pakistan with wide inconsistencies that exist in the country's road infrastructures, available transport systems, and surgical functionality of local hospitals. Public facility databases are not comprehensive, do not include private facilities, and do not monitor for functionality. Reliable GIS resources required to calculate accurate travel times with Pakistan's unique and diverse terrain and infrastructure were not found.

Our novel and unique community-centered approach to estimate access through a 2 stage randomized cluster survey was therefore deemed a more accurate approach to estimate the true available access. It involved a robust sampling strategy and real community based data that allowed us to get accessibility estimates based on the local knowledge about travel times and the preferred surgical facilities in the community. However, there were significant limitations with this approach. Since we were unable to cross check the functionality of facilities and accuracy of travel time, we may have over- or under- estimated surgical care access. We tried to mitigate the risk of response bias by hiring local health workers who were trained and were able to explain procedures in layperson terms in local languages. We therefore believe the effect of inaccurate response is minimal and our approach estimates the true access better than a GIS based approach would. Additional validation studies are needed; we are currently working on a comparison of our methodology with GIS generated access data and plan to publish the findings in a separate paper. Secondly, the study was more cost and energy intensive than GIS based studies, and required a large team of health workers given the language diversity across the country. Consistency was maintained in the data monitoring team to ensure standardization in data collection. In comparison to the national household consumption quintiles, we had a higher representation of households in lower quintiles. This may be attributed to the difference in data collection tools. Nevertheless, the sample was representative of the wide and diverse geography of the country.

Our study is the first to use a household survey to estimate surgical access in LMICs. We plan to validate our approach by comparing our findings with a GIS based approach. The study also provides unique insights into the dynamics of surgical care accessibility in Pakistan. The poor estimated access to surgical care in rural areas and in the provinces of Balochistan, KPK, and Sindh predisposes the population in these areas to preventable mortality and morbidity from injuries and emergency general surgery conditions. The pilot implementation of the DCP3 essential surgical packages at secondary level hospitals can lead to better surgical accessibility [32]. During the scale-up of surgical services, the quality and safety of surgical services should not be compromised and efforts to reduce burden of surgical conditions should also be made. Taking a lesson from Zambia, routine household surveys such as Demographic Health Survey (DHS) or Household Integrated Economic Survey (HIES)/ Pakistan Standards of Living Survey (PSLM) conducted every 2–3 years, should be used to study population surgical access in the future [17, 22, 33]. Following the recommendations of the Utstein meeting on surgical care indicators, the government, in collaboration with the private sector, needs to develop a database of facilities with Bellwether capacity and a locally accurate GIS methodology to estimate population surgical care access as per the consensus [34].

## Supporting information

**S1 Text. Annex 1 questionnaire.**
(PDF)

**S1 Table. Supplemental table with design effects (DEFF for accessibility estimates).**
(DOCX)

## Acknowledgments

People conducting fieldwork, Hira Zuberi, Lisa Nussbaum

## Author Contributions

**Conceptualization:** Muhammad Nabeel Ashraf, Katherine Albutt, Asad Latif, John G. Meara, Lubna Samad.

**Data curation:** Muhammad Nabeel Ashraf, Irum Fatima, Ammar Ali Muhammad, John G. Meara.

**Formal analysis:** Muhammad Nabeel Ashraf, Ammar Ali Muhammad, Manon Pigeolet, Asad Latif, Lubna Samad.

**Funding acquisition:** John G. Meara, Lubna Samad.

**Investigation:** Muhammad Nabeel Ashraf, Asad Latif, John G. Meara, Lubna Samad.

**Methodology:** Muhammad Nabeel Ashraf, Ammar Ali Muhammad, Katherine Albutt, Asad Latif, John G. Meara, Lubna Samad.

**Project administration:** Muhammad Nabeel Ashraf, Irum Fatima, Ammar Ali Muhammad, John G. Meara, Lubna Samad.

**Resources:** Irum Fatima, Lubna Samad.

**Supervision:** Irum Fatima, Asad Latif.

**Validation:** Asad Latif, John G. Meara, Lubna Samad.

**Visualization:** Muhammad Nabeel Ashraf.

**Writing – original draft:** Muhammad Nabeel Ashraf.

**Writing – review & editing:** Muhammad Nabeel Ashraf, Irum Fatima, Ammar Ali Muhammad, Manon Pigeolet, Asad Latif, John G. Meara, Lubna Samad.

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
