## [Decision Letter · Decision Letter 0]

28 Feb 2023

PGPH-D-23-00001

Access to Surgical Care in Pakistan; Findings of a National Household Survey

Dear Dr. Ashraf,

Thank you for submitting your manuscript to PLOS Global Public Health. After careful consideration, we feel that it has merit but does not fully meet PLOS Global Public Health’s publication criteria as it currently stands. Therefore, we invite you to submit a revised version of the manuscript that addresses the points raised during the review process. In particular, I think it is essential to address how confident we can be that laypeople can accurately know the availability of surgical care in any particular insitution. 

We look forward to receiving your revised manuscript.

Kind regards,

M. Dylan Bould

Academic Editor

Journal Requirements:

1. Please send a completed 'Competing Interests' statement, including any COIs declared by your co-authors. If you have no competing interests to declare, please state "The authors have declared that no competing interests exist". Otherwise please declare all competing interests beginning with the statement "I have read the journal's policy and the authors of this manuscript have the following competing interests:"

3. Please provide separate figure files in .tif or .eps format only and remove any figures embedded in your manuscript file. Please also ensure that all files are under our size limit of 10MB.

4. Fig 1: please (a) provide a direct link to the base layer of the map (i.e., the country or region border shape) and ensure this is also included in the figure legend; and (b) provide a link to the terms of use / license information for the base layer image or shapefile. We cannot publish proprietary or copyrighted maps (e.g. Google Maps, Mapquest) and the terms of use for your map base layer must be compatible with our CC-BY 4.0 license. 

Additional Editor Comments (if provided):

Interesting choice of household survey to collect information on this topic. I agree with the comments by one of the reviewers regarding clarification about what data is self reported. How well do lay people in the community know whether they have reliable access to a particular type of surgical procedure?

Could you please include the full interview tool as an appendix please?

The numbers in table 2 could be reduced to 2 decimal places to improve readability , or otherwise would me more easily read if converted to percentages with one decimal place.

Please don't use abbreviations like "C-section", however obvious the abbreviation may be to you, without first spelling the abbreviation out in full. Another example is "LMIC" which could be taken to mean lower-middle income country or low- and middle-income country. There are other examples in the text (DALY).

In the abstract you write "Pakistan is an LMIC in South Asia". It is not specified whether you mean lower-middle income country or low- and middle- income country. Please see these publications which caution against the ubiquitous and unreflective use of the term "LMIC"

https://www.ncbi.nlm.nih.gov/pmc/articles/PMC9185671/#:~:text=The%20use%20of%20low%2Fmiddle,the%20countries%27%20Gross%20National%20Income

https://www.ncbi.nlm.nih.gov/pmc/articles/PMC9185389/

At best, this term tends to over-generalize.

"Sub-Saharan" should be "sub-Saharan"

Perhaps "The Lancet Commission on Global Surgery (LCoGS) estimates 5 billion people lack access to timely, safe, quality, and affordable surgical care" should be written in the past tense, as it is now 8 years old and indeed the world population is significantly higher now than it was in 2015, and it's likely that more than 5 billion lack access to safe, timely and affordable surgical care now.

In the discussion you write "While population access to C-sections is found to be above 80%, population access to Laparotomy, OFR, and specialized surgery is suboptimal." I note that 80% access to Cesarean section is still not "optimal" even if it meets your agreed target.

The discussion section is rather long and could be made more concise.

Your link for the data does not seem to work - can you check this please?

Reviewers' comments:

Reviewer's Responses to Questions

**Comments to the Author**

1. Does this manuscript meet PLOS Global Public Health’s publication criteria? Is the manuscript technically sound, and do the data support the conclusions? The manuscript must describe methodologically and ethically rigorous research with conclusions that are appropriately drawn based on the data presented.

Reviewer #1: Yes

Reviewer #2: Yes

2. Has the statistical analysis been performed appropriately and rigorously?

Reviewer #1: Yes

Reviewer #2: Yes

3. Have the authors made all data underlying the findings in their manuscript fully available (please refer to the Data Availability Statement at the start of the manuscript PDF file)?

Reviewer #1: Yes

Reviewer #2: Yes

4. Is the manuscript presented in an intelligible fashion and written in standard English?

Reviewer #1: Yes

Reviewer #2: Yes

5. Review Comments to the Author

Reviewer #1: "Poor surgical care service availability at secondary-level hospitals that are supposed to be the

front line for emergency surgical services is one of the main reasons for the rural disparity, Despite being designated as centers for essential surgery, surveys of rural secondary-level

hospitals have shown that up to 71% do not have an anesthetist", I am curious to know if there are any bridging courses available for medical officers to administer spinal anesthesia for acute emergencies like a c section.

To mitigate the constraint of specialist providers , the de facto standard of care in many nations is task-sharing of anesthesia provision, which is estimated to occur in as many as 119 countries, across all World Bank income groups. In India, medical officers, who are non-specialist medical graduate physicians, have a precedent of delivering anesthesia care in rural hospitals. Medical officers are incentivized to work in non-urban healthcare settings in a number of ways inclusive of subsidized medical education at government medical colleges or as a mandatory requirement to fulfil prior to postgraduate specialization. Just curious to know if such a program is available in Pakistan, as many of the described staff shortages can be phased out

Reviewer #2: This is a very good and timely study measuring timely access to surgical care using a household survey, in a lower middle-income country that is hypothesised to have poor access. As rightly discussed one would wonder why not Geospatial mapping for this study but the authors explain the limitations of GIS studies in such settings. My concerns are mainly -seeking clarity with the methodology, and the interpretation of results.

1. Methodology: Would a ordinary member of the community be expected to know whether services for laparotomy and fractures are available – or was this information matched between information from the tehsils. This could be better detailed in the methodology. This will determine whether the data is fully accurate or based on the perspectives of the people interviewed, rather than availability of services.

2. Results – reported access was lower for laparotomy, then caesarean section – was there any way of ensuring that the patients clearly understood the definitions?

3. Table 4 – this is very interesting. Again is this reported? Is it accurate. Could it be assumptions. Was there an attempt to verify. In many LMIC’s with lower levels of education, communities may not always know what services are available even in their local hospitals, especially when it comes to specialised surgical services.

4. Discussion – line 1, I would recommend that you add – ‘according to a household survey’.

5. Discussion paragraph 2: Access to essential surgery does not equal access to specialist surgeons. It is very common is LMICs that surgeries are not done by specialists, even complex and complicated surgeries. Even in tertiary hospitals the primary surgeons is usually a registrar or resident who calls for help when in need of a specialist. This is not surprising in the context of an LMIC. This must be explained in the context of this country and not generalised so that we know who is actually doing the operations.

6. It does appear that the solution to access may not be increasing volume – if the country needs to go from 89/100 000 to 5000/100 000. The feasibility of the recommendation by the Lancet Commission on Global Surgery needs to stated with caution. Reducing the need for surgery is a strategy that may go hand in hand with increasing the surgical volume and surgeons.

7. Discussion final paragraph: Even with the proposed institutionalisation of indicators the interpretation of data needs to be taken in context of the local setting. The concern with pushing for a focus on numbers is that generalisations that lead to misinterpretation of data can be made – e.g. what does 74.8% compared to benchmark of 80% really say to us about access to surgical care, especially if we do not know who is operating (specialist vs non-specialist) and what the outcomes of the patients are (dead/alive and healthy/severe complications). Please expand more on what we really learn from this finding – and why the idea of a national indicator would improve access.

Recommendation: Accept with minor edits.

6. PLOS authors have the option to publish the peer review history of their article (what does this mean?). If published, this will include your full peer review and any attached files.

**Do you want your identity to be public for this peer review?** For information about this choice, including consent withdrawal, please see our Privacy Policy.

Reviewer #1: **Yes: **Wesley Rajaleelan

Reviewer #2: No

---

## [Decision Letter · Decision Letter 1]

17 May 2023

PGPH-D-23-00001R1

Access to Surgical Care in Pakistan; Findings of a National Household Survey

Dear Dr. Ashraf,

Thank you for submitting your manuscript to PLOS Global Public Health. After careful consideration, we feel that it has merit but does not fully meet PLOS Global Public Health’s publication criteria as it currently stands. Therefore, we invite you to submit a revised version of the manuscript that addresses the points raised during the review process.

We look forward to receiving your revised manuscript.

Kind regards,

M. Dylan Bould

Academic Editor

Journal Requirements:

Additional Editor Comments (if provided):

For me, there is a key concern that has not been fully addressed. Unless I'm mistaken, this study uses a novel method of assessing access to surgery - it seems that household surveys are well established for estimating population needs/burden of disease, but not access to surgical care. This method may or may not be superior to other methods, but does not seem to be validated or triangulated against any other method. While this does not in my mind necessarily preclude publication, it needs to be dealt with more clearly and earlier in the manuscript, and emphasized.

A concern raised, as noted above, was whether the participants were able to accurately evaluate their distance from a center which could perform a Bellwether procedure. You responded that the survey actually collected data on which institution they would choose, and how long they anticipated it would take them to get there. Therefore, this data is assessing perceived accessibility rather than actual accessibility. However, throughout the paper you still refer to access rather than perceived access.

I would like to confirm - did you in any way assess how well perceived travel times match actual travel times?

Also, did you in any way confirm that the institutions that respondents chose were actually able to provide the Bellwether procedure. It is possible that the participants could have suggested an institution that actually was not able to provide the procedure - but was this captured in any way? For this reason the actual access could possibly be quite different to the perceived access.

"The study team reviewed the responses on a regular basis for plausibility" is still too vague.

I think we need to have more clarity on these two questions, as the validity of your findings rests largely on this.

I can appreciate the limitations of geographic information system (GIS) data in the Pakistan context, and the desire to find a more accurate method of assessing access to surgery. It could be argued that it remains unclear to what extent household survey data relates to actual access to surgery, and how its accuracy compares to the accuracy of GIS data. I don't think that this comes across on your paper.

Reviewers' comments:

Reviewer's Responses to Questions

**Comments to the Author**

1. If the authors have adequately addressed your comments raised in a previous round of review and you feel that this manuscript is now acceptable for publication, you may indicate that here to bypass the “Comments to the Author” section, enter your conflict of interest statement in the “Confidential to Editor” section, and submit your "Accept" recommendation.

Reviewer #1: All comments have been addressed

Reviewer #3: (No Response)

2. Does this manuscript meet PLOS Global Public Health’s publication criteria? Is the manuscript technically sound, and do the data support the conclusions? The manuscript must describe methodologically and ethically rigorous research with conclusions that are appropriately drawn based on the data presented.

Reviewer #1: Yes

Reviewer #3: Yes

3. Has the statistical analysis been performed appropriately and rigorously?

Reviewer #1: Yes

Reviewer #3: Yes

4. Have the authors made all data underlying the findings in their manuscript fully available (please refer to the Data Availability Statement at the start of the manuscript PDF file)?

Reviewer #1: Yes

Reviewer #3: No

5. Is the manuscript presented in an intelligible fashion and written in standard English?

Reviewer #1: Yes

Reviewer #3: Yes

6. Review Comments to the Author

Reviewer #1: The authors have addressed my concerns this this revision

Reviewer #3: The link to data is not working.

https://dataverse.harvard.edu/privateurl.xhtml?token=bc6cbda9-c33c-4394-99166f893f3ba593

7. PLOS authors have the option to publish the peer review history of their article (what does this mean?). If published, this will include your full peer review and any attached files.

**Do you want your identity to be public for this peer review?** For information about this choice, including consent withdrawal, please see our Privacy Policy.

Reviewer #1: **Yes: **Wesley Rajaleelan

Reviewer #3: **Yes: **Kennedy Misso

---

## [Decision Letter · Decision Letter 2]

1 Sep 2023

PGPH-D-23-00001R2

Estimating Access to Surgical Care: A Community Centered National Household Survey from Pakistan

Dear Dr. Ashraf,

Thank you for submitting your manuscript to PLOS Global Public Health. After careful consideration, we feel that it has merit but does not fully meet PLOS Global Public Health’s publication criteria as it currently stands. Therefore, we invite you to submit a revised version of the manuscript that addresses the points raised during the review process.

We look forward to receiving your revised manuscript.

Kind regards,

Hassan Haghparast Bidgoli

Academic Editor

Journal Requirements:

Reviewers' comments:

Reviewer's Responses to Questions

**Comments to the Author**

1. If the authors have adequately addressed your comments raised in a previous round of review and you feel that this manuscript is now acceptable for publication, you may indicate that here to bypass the “Comments to the Author” section, enter your conflict of interest statement in the “Confidential to Editor” section, and submit your "Accept" recommendation.

Reviewer #3: All comments have been addressed

2. Does this manuscript meet PLOS Global Public Health’s publication criteria? Is the manuscript technically sound, and do the data support the conclusions? The manuscript must describe methodologically and ethically rigorous research with conclusions that are appropriately drawn based on the data presented.

Reviewer #3: Yes

3. Has the statistical analysis been performed appropriately and rigorously?

Reviewer #3: Yes

4. Have the authors made all data underlying the findings in their manuscript fully available (please refer to the Data Availability Statement at the start of the manuscript PDF file)?

Reviewer #3: Yes

5. Is the manuscript presented in an intelligible fashion and written in standard English?

Reviewer #3: Yes

6. Review Comments to the Author

Reviewer #3: Introduction

First paragraph

the reference cited reported a range of 28-32%. And the author referring to results of another author. It may be corrected to “Shrime reports are health care providers' estimates, which they reported to be 28-32%”.

To suffice English language in publications you may need to change/rephrase the sentence beginning with the majority of these…… “Injuries (77%), maternal and neonatal conditions (14%), and gastrointestinal conditions contributed to the majority of preventable mortality”.

Methodology

Specify the duration of interruption from a Covid-19 pandemic.

Data collection- Identify members of the household who was/were interviewed. Some questions on household consumption may be sensitive in some cultures or require a leading member of the household.

Results

After Table 1, the second table needs rephrasing as the numbers are too close. It may read C-sections procedures had the highest access of 88.7%, followed by OFR at 81% and laparotomy at 79%"

In table two, the column with All procedures, does it include specialized and Bellwether procedures? If so, it needs to be specified in such rather than All procedures.

Opening statements before Table 3- To my understanding this was a household survey, avoid reporting population access. And provide a summary of the table. Rephrase "Highest access to Bellwether procedures is observed in Islamabad CT (92.9%), and Punjab (81%). In contrast, Balochistan had the lowest rate (28.6%) (Table 2).

You may choose one method of statistical significance, either a CI or P value.

Discussion

Second last paragraph- This sounds like a study limitation

Good work to authors. Some adjustments will polish this work even better.

7. PLOS authors have the option to publish the peer review history of their article (what does this mean?). If published, this will include your full peer review and any attached files.

**Do you want your identity to be public for this peer review?** For information about this choice, including consent withdrawal, please see our Privacy Policy.

Reviewer #3: **Yes: **Kennedy Kisengo Misso

---

## [Editor Report · Decision Letter 3]

26 Sep 2023

PGPH-D-23-00001R3

Estimating Access to Surgical Care: A Community Centered National Household Survey from Pakistan

Dear Dr. Ashraf,

Thank you for submitting your manuscript to PLOS Global Public Health. After careful consideration, we feel that it has merit but does not fully meet PLOS Global Public Health’s publication criteria as it currently stands. Therefore, we invite you to submit a revised version of the manuscript that addresses the points raised during the review process.

Editor Comments:

Thanks for addressing the issues raised by the reviewers. Please address the following issues:

- Please explain how informed consent was obtained from the participants and the institutions where ethical approval were obtained.

- You have reported that "We assumed an intra-cluster correlation coefficient (ICC) of 1/6". What is 1/6? ICC can have a value between 0-1 and commonly is between 0.01 and 0.02 . Please correct this.

We look forward to receiving your revised manuscript.

Kind regards,

Hassan Haghparast Bidgoli

Academic Editor
---

## [Editor Report · Decision Letter 4]

18 Oct 2023

Estimating Access to Surgical Care: A Community Centered National Household Survey from Pakistan

PGPH-D-23-00001R4

Dear Dr. Ashraf,

We are pleased to inform you that your manuscript 'Estimating Access to Surgical Care: A Community Centered National Household Survey from Pakistan' has been provisionally accepted for publication in PLOS Global Public Health.

Best regards,

Hassan Haghparast Bidgoli

Academic Editor
